# Seasonal Forest Changes of Color and Temperature: Effects on the Mood and Physiological State of University Students

**DOI:** 10.3390/ijerph20146338

**Published:** 2023-07-10

**Authors:** Eunjin Kim, Hwayong Lee

**Affiliations:** 1Department of Forest Therapy, Chungbuk National University, Cheongju 28644, Republic of Korea; kimej@chungbuk.ac.kr; 2Department of Forest Science, Chungbuk National University, Cheongju 28644, Republic of Korea

**Keywords:** forest color, university students, mood state, urban forest planning

## Abstract

In this study, we attempted to analyze the effect of color and temperature changes in the forest environment over time on the mood and physiological state of university students. The survey was conducted four times considering forest changes such as new leaf appearance and growth, autumn leaf changes, and fallen leaves. The participants’ moods and physiological states were first evaluated in an indoor environment; a second evaluation was conducted after contact with the forest. The color visual information of the forest environment was analyzed through color extraction from photographs taken each survey day. The participants’ moods and physiological states were measured using the Korean Profile of Mood States-Brief and a heart rate variability measuring device, respectively. Changes in the forest experience according to the season had an effect on university students’ mood states. In particular, the effects of the spring forest experience included the relaxation of tension and the activation of vigor. This result is considered to be influenced by factors such as the season’s temperature and the green color, which is predominant in the spring forest. However, no physiological changes were found in the participants according to each season. The results of this study can lead to greater consideration of the role of color in urban forest planning for universities and other public spaces.

## 1. Introduction

High levels of stress and anxiety have been routinely reported among college and university students [1,2]. These increased levels of stress and anxiety have been attributed to a heavy academic burden, sleep problems, competition with peers, concerns about the future, and, in some countries, financial concerns [3,4]. Interest has grown in the campus environment’s role in reducing mental health risks and improving the psychological well-being of university students [5]. One topic that has attracted particular attention is how the creation of green spaces on campus can potentially benefit students’ psychological health [6]. There is an example of using a school forest as a solution to adolescent behavior problems [7].

Natural landscapes are known to relieve psychological stress in humans [8]. The natural environment has fewer straight lines compared to artificial environments; the shape of trees, the visual diversity of the landscape, and the naturalness of monotonous colors can all have a restorative effect on individuals [9]. Human contact with nature offers the advantage of a positive emotional state with reduced stress and increased relaxation, which can contribute greatly to mental recovery [8,10,11,12]. Nature has been proven to positively influence human psychology [13,14]. Therefore, contact with the natural environment could alleviate the mental fatigue suffered by many people in modern society [15]. Forests have been shown to contribute to improving human emotional health [16]. Furthermore, forest activities can reduce hostility and depression and increase vigor in human interactions [17]. Moreover, emotional states such as tension and anxiety, depression and dejection, anger and hostility, energy, confusion, and fatigue can be improved through forest activities, thereby creating a sense of psychological relaxation [18].

Mood is defined as a mild, pervasive, generalized emotional state that is perceived subjectively by an individual [19]. It affects not only our overall sense of well-being but also our behavioral patterns and perceived health. Measuring mood states can determine the effect of interference on mood states and disorders [20,21,22]. The “Profile of Mood States” (POMS) questionnaire is widely used in clinical practice when examining issues relating to psychotherapy, psychology, and physical problems [23].

Heart rate variability (HRV) is known as a quantitative and objective measure of the autonomic nervous system’s ability to cope in stressful situations [24,25,26,27]. HRV measurement is widely used to obtain information on stress-related diseases because it is a simple, non-invasive method, and the results can be obtained immediately [28,29]. HRV is one of many pathological indicators related to health; high HRV is associated with a healthy state, while low HRV is associated with a pathological state [30]. Therefore, participants’ physiological states can be estimated using their HRV following the forest experience.

Interest in the seasonal color design of plants is increasing [31], and images of ornamental plants and psychophysiological effects on humans have been investigated for use in urban greening projects [32]. In addition, each plant community, such as single-layer woodland, tree-shrub-grass composite woodland, and tree-grass composite woodland, is known to affect human physiological recovery and emotional response [33]. Previous studies have confirmed the health benefits of green nature [34,35,36,37], maple forests [38], and seasonal (spring and autumn) forest contact [39]. These results suggest that changes in forest conditions perceptible through human senses can affect human psychological and physiological changes.

In this study, we used the K-POMS-B and HRV to investigate the difference in the moods and physiological states of university students who experienced seasonal differences in color and temperature.

## 2. Materials and Methods

### 2.1. Research Site and Survey Period 

The study site was the urban forest area of the Chungbuk National University campus, in Cheongju city, Chungchungbuk-do province, Republic of Korea. The forest covers an area of 25,711 m^2^, with a 4:6 ratio of deciduous broad-leaved trees to evergreen conifers in the upper layer of the forest. The middle layer, located in the field of view of pedestrians, consists of deciduous trees, such as *Prunus serrulata*, *Quercus acutissima*, *Liriodendron tulipifera*, *Styrax japonicus*, and *Dalbergia hupeana*. The forest is located close to university housing and local residential areas and has a walking trail that is accessible 24 h a day, year round. Students and local residents use the forest as a place for rest and recreation (Figure 1). Dead branches that can pose a danger to forest users are continually removed, and disease and pest management are carried out to maintain a healthy forest.

In consideration of seasonal forest changes, such as leaf opening, autumn leaf changes, and fallen leaves, the survey was conducted four times: 11 November 2021 (first survey, autumn), 12 December 2021 (second survey, winter), 27 April 2022 (third survey, spring), and 15 June 2022 (fourth survey, summer). Each survey was conducted on a sunny day from 11:00 to 15:00. The average, maximum, and minimum temperatures for each survey day are shown in Table 1. The temperature records of the meteorological station closest to the research site (1.7 km) were used. The temperature of the indoor environment was maintained at around 25 °C throughout the year.

### 2.2. Changes in Forest Color Distribution by Season

The forest colors were recorded using a digital camera (Nikon, D3200, Tokyo, Japan). The photos taken each survey day were compared by extracting the color of the forest and indoor environment using the color extraction feature of Cool PHP Tools (www.coolphptools.com (accessed on 11 May 2022). The indoor environment was a student research office.

### 2.3. Characteristics of the Participants and Survey Method

The study objective was to compare changes in human mood according to seasonal forest environment experiences. Differences in the main colors of the seasonal forest environment at the research site were compared, and participants’ changes in mood state were compared after short walks in the indoor environment and seasonal forest environment. The survey was conducted once for each season.

The number of study participants for each survey day was 22 for the first (male: 8, female: 14), 22 for the second (male: 7, female: 15), 28 for the third (male: 12, female: 16), and 30 for the fourth (male: 13, female: 17) (Table 2). The study participants were students attending C University in Cheongju City, and were recruited on a first-come, first-served basis through school bulletin boards and SNS announcements. Participants were recruited every survey day without repeating the same number of participants. The participants had no major physical or mental problems. This study was conducted after prior deliberation and approval by the Bioethics Committee of Chungbuk National University (IRB approval: CBNU-202110-HR-0152).

During the experiment, participants entered the laboratory and sat in a chair. They were briefed on the experimental procedures and instructions. After participants sat in the chair for 10 to 20 min and took a break to adjust to the laboratory environment, their moods and physiological states were initially measured. For the second measurement, the participants relocated to the forest, walked for 10 to 20 min as in the laboratory, and focused their eyes on the experimental measurement site located within the forest. At this point, their moods and physiological states were re-measured. 

The K-POMS-B measurement index used in this study comprised six subscales: “tension”, “anger”, “depression”, “fatigue”, “confusion”, and “vigor” (Table 3). A total of 30 questions (5 questions for each subscale) were scored on a 5-point Likert scale ranging from 1 point for “not at all” to 5 points for “very much so”.

The POMS has also been utilized to assess transient and distinct mood states [40]. The original version comprises 65 questions scored on a 5-point scale, covering the period “last week including today,” and assesses levels of depression, tension-anxiety, anger-hostility, fatigue-inertia, vitality-activity, and confusion-embarrassment [19]. The POMS-B is a shorter, easier to use version of the original POMS and consists of 30 adjectives describing the emotions and moods respondents may have experienced [41]. The POMS-B is used in several countries including the United States and Republic of Korea, where its use has been verified [42]. The questionnaire employed in this study was created with the K-POMS-B (Korean Version of Profile of Mood States-Brief).

Changes in participants’ physiological states were compared using their HRV after the forest experience for each survey day. The participants measured their heart rates at their fingertips for 2.5 min using a heart rate variability measuring device (uBioMacpa, Biosense Creative, Seoul, Republic of Korea) while sitting and resting in the indoor environment and in the forest on each survey day. The low frequency component (LF), high frequency component (HF), and LF/HF ratio were measured using the heart rate variability measuring device. LF indicates the activity of the sympathetic nervous system and can decrease due to a lack of energy, fatigue, etc., and HF indicates the activity of the parasympathetic nervous system and can decrease due to chronic stress, aging, etc. [43]. The LF/HF ratio indicates the degree of balance of the autonomic nervous system, and 0.5 to 2.0 is considered a balanced state [44].

The collected data were analyzed using the SPSS statistics program, version 18. To verify the homogeneity of the study participants, an analysis of variance (ANOVA) was performed on the mood state results in the indoor environment on each of the survey days. A paired *t*-test was conducted to examine the differences in mood state between the forest and indoor environment for each survey day. ANOVA analysis and Bonferroni correction were performed to compare mood and physiological states for each season.

## 3. Results

### 3.1. Color Distribution in the Forest by Season

A forest provides a wide range of visual information, such as flowering, leaf opening, duration of flowers and leaves, and autumn foliage, depending on the types of plants that comprise the forest. Correspondingly, urban trees offer physical and aesthetic benefits related to the presence or absence of leaves [45]. In the present study, the color composition of the study site varied across each of the survey days. The first survey day corresponded to autumn, when the leaves change to autumn foliage. At this time, red, yellow, and orange colors not commonly seen during other times of the year appeared in the forest. The second survey day occurred during early winter when the leaves of deciduous broad-leaved trees fall. At this time, the forest was composed of gray and black colors, similar to the indoor environment but darker. The third survey day fell during the spring season when the leaves opened, and the fourth survey day occurred during the summer when the foliage grew vigorously. On both the third and fourth survey days, the forest was mainly composed of green colors. On the third survey day in particular, the three colors of very dark gray, lime green (Hex code: #304830, #486048), and mostly black (Hex code: #181818) accounted for more than 90% of the forest color profile. At this time, the color composition was very simple compared to the other survey days. Saturation and color change stimulate the eyes to remove negative thoughts and help individuals pursue positive life values. Moreover, it has been proven that color therapy using saturation and color change can improve the quality of life of stroke patients and their caregivers [46]. The rates of the main colors extracted from the forest and indoor environment for each survey are shown in Figure 2.

### 3.2. Homogeneity Verification of the Research Participation Group

The K-POMS-B was used to measure each participating group’s mood state in an indoor environment with no discernible difference among survey days. The homogeneity of each participating group was confirmed through ANOVA analysis. Analysis of the K-POMS-B sub-factors revealed no difference in tension, anger, depression, fatigue, and confusion for each survey day, but there was a significant difference in vigor (F = 3.262, *p* < 0.05) (Table 4). Therefore, it can be inferred that tension, anger, depression, fatigue, and confusion among each sub-factor of the K-POMS-B had no effect according to the survey day.

### 3.3. University Students’ Mood State Changes after Forest Contact

In this study, the K-POMS-B was used to discern the effect of forest contact on university students’ mood state on each survey day. When the POMS-B test tool was developed, the Cronbach’s α coefficient was 0.63 to 0.96. The Cronbach’s α coefficient in this study was 0.89, indicating that there was no problem in determining the reliability of the measurement factor. 

University students’ mood states determined by the K-POMS-B showed differences after forest contact and in the measurements recorded in the indoor environment on each survey day. On the first survey day, the values of “tension” (t = 2.611, *p* < 0.05) and “fatigue” (t = 3.813, *p* < 0.05) were significantly decreased after forest contact, as compared to the indoor environment. In addition, although not significant on this survey day, unlike during the other three survey days, the value of the “anger” factor increased after forest contact compared to the measurements recorded in the indoor environment, and the value of the “vigor” factor decreased after forest contact compared to the indoor environment. The pattern on the second survey day was generally the same as that on the third and fourth survey days. Only the value of the “fatigue” (t = 2.618, *p* < 0.05) factor decreased significantly after forest contact compared to the indoor environment, while the value of the “vigor” (t = −3.226, *p* < 0.01) factor increased significantly. All sub-factors of university students’ K-POMS-B on the third survey day showed statistically significant differences with the indoor environment after forest contact. At this time, the values of “tension” (t = 4.972, *p* < 0.001), “anger” (t = 4.492, *p* < 0.001), “depression” (t = 3.490, *p* < 0.01), “fatigue” (t = 4.715, *p* < 0.001), and “confusion” (t = 3.894, *p* < 0.01) decreased after forest contact compared to the indoor environment. Conversely, the value of “vigor” (t = −5.590, *p* < 0.001) increased after forest contact as compared to the levels recorded in the indoor environment. On the fourth survey day, all POMS-B subscales showed the same pattern as on the third survey day, with only the factors of “tension” (t = 3.093, *p* < 0.01), “anger” (t = 2.448, *p* < 0.05), and “fatigue” (t = 2.285, *p* < 0.05) showing statistically significant decreases after forest contact compared to the indoor environment (Figure 3).

### 3.4. University Students’ Physiological States Using HRV Changes after Forest Contact

In this study, HRV was used to compare the difference in university students’ physiological states after the indoor environment (before the experiment) and forest contact on each survey day. After the forest experience on the third survey day, “LF/HF” was found to be higher for the forest experience than the indoor environment (t = −2.073, *p* < 0.05), and no statistical difference was found between LF and HF. No statistical difference was found between LF, HF, and LF/HF after the forest experience in the first, second, and fourth survey days (Table 5).

### 3.5. Differences in Sub-Factors of Mood State in Indoor and Seasonal Forest Environments

Differences between mood state sub-factors in indoor and seasonal forest environments were analyzed using a one-way ANOVA, and post-hoc tests were analyzed using Bonferroni’s correction.

A statistically significant difference was found in the mood state sub-factor “tension”. The forest environment experience on the first, third, and fourth survey days was found to be less tense than that of the indoor environment according to the post-hoc test. No significant difference was found for “anger” by season in the post-hoc test. A statistically significant difference was found for “depression”, but no significant difference was found by season in the post-hoc test. There was a significant difference in “fatigue” between the indoor and forest environments by each survey day. The forest experience on the third survey day showed lower “fatigue” than the indoor environment. In particular, the forest experience was confirmed to be significantly lower than the indoor environment on the first, second, and third survey days of investigation in the post-hoc test. There was a statistically significant difference in “confusion”, but no significant difference was found by season in the post hoc-test. A significant difference was found for “vigor” between the indoor and forest environments by each survey day; in the post-hoc test, the forest experience of the second and third survey days was found to show higher “vigor” than the indoor environment. To summarize, “tension” showed statistically significant results in the forest experience of the first, third, and fourth survey days and had the greatest effect in the third survey day. “Fatigue” showed statistically significant results in the forest experience of the first, second, and third survey day, and a significant effect was found in the forest experience of the first survey day. “Vigor” showed the greatest effect in the forest experience of the second survey day. The third survey day was effective for “tension”, “fatigue”, and “vigor” (Table 6).

### 3.6. Differences in Physiological State Sub-Factors in Indoor and Seasonal Forest Environments

In this study, university students’ physiological sub-factors for differences in indoor and seasonal forest environments were analyzed using a one-way ANOVA. The LF, HF, and LF/HF responses did not show statistically significant differences between seasons (Table 7).

## 4. Discussion

Contact with forests, parks, and gardens plays an important role in nurturing human well-being [47,48,49,50,51]. The proximity of a natural park helps to create a pleasant and stable life and affects people’s psychological and physiological recovery [52,53,54]. Seasonal forest experiences showed differences in positive changes in university students’ mood state. 

A forest provides a wide range of visual information, such as flowering, leaf opening, duration of flowers and leaves, and autumn foliage, depending on the types of plants comprising the forest. Correspondingly, urban trees offer physical and aesthetic benefits related to the presence or absence of leaves [45]. Among the seasonal characteristics of the research site, the color differences of vegetation in the forest environment were prominent. Throughout the experiment period, the indoor environment maintained the same conditions, such as a 25 °C temperature and gray, grayish yellow, and blue color compositions. Saturation and color changes are also known to affect quality of life [46]. In this study, the differences in the forest environment colors were clearly visible across the seasons. Therefore, differences were found in the visual information of the forest environment by season, and in human mood and physiological state, respectively.

Tension was lower in the forest environment compared to the indoor environment; in particular, the effect of relaxation was observed after participants experienced the forest environment in spring, fall, and summer. Exercise in green spaces has been reported to play an important role in managing and supporting mental health recovery by lowering tension [55]. This was further demonstrated in the present study, as university students’ tension was demonstrably lower following forest contact, as compared to measurements recorded in the indoor environment in spring and summer, when green was the main color. However, tension was also lower after forest contact in autumn, when red, yellow, and orange colors were predominant. In a study by Song et al. [56], the effect of walking in an urban forest in winter, when leaves turned red or yellow and the average temperature was 13.8 °C, was proven to lower tension. In the present study, when the leaves were mainly colored red, yellow, and orange, the temperature at the meteorological station closest to the research site was 10.8 °C [57]. This result is considered to be very close to that of Song et al. [56]. Therefore, the visual experience provided by the green forest environment in spring and summer and the mild temperature experience of the autumn forest environment are believed to have successfully lowered the study participants’ tension level as compared to the indoor environment.

In this study, contact with the forest in spring and summer, when green is the main color on display, lowered university students’ anger level. Green colors are also known to be effective in reducing anger in pregnant women [58]. Although the results were not statistically significant, in autumn, when red colors were prominent, university students’ anger increased markedly as compared to other seasons. A study by Akers et al. [14] also concluded that anger levels increased in healthy male participants after exercise in the red visual field condition; therefore, it is thought that exposure to red colors can increase anger. However, flowers that present warm colors, such as orange, yellow, and red, can also evoke uplifting emotions and have a better positive impact [59]. It is believed that humans perceive natural images, such as trees and flowers, differently. Therefore, even the same color could have a different effect on a person’s mood depending on the substance comprising that color.

Vision loss and depression have been documented as being related [60], while color intensity was found to be significantly correlated with depression in a study of inpatients and outpatients [61]. In the present study, depression in university students was shown to be significantly lower only in spring. More than 90% of the color profiles presented comprised seven colors in the indoor environment: two colors in spring, four colors in summer, and five colors in autumn and winter. Spring was the simplest, with one color (#304830) accounting for 52.75% of the color profile. The composition of colors in the environment is thought to be closely related to depression. More research is needed to further investigate the relationship between forests’ color composition and depression.

Confusion in university students was lower during forest contact than indoors in all seasons and showed significant results in spring. In a study by Takayama et al. [62], a 15-min period spent in the forest in summer lowered the confusion level of male university students as compared to results recorded in the city, which matched the result in the present study. In a study by Bielinis et al. [63], in both spring and winter seasons, the confusion level of young adults was lower in the forest than indoors, with a significant contrast recorded in winter. While forest contact is thought to reduce confusion, no association was found in the present study between color change and reduction of confusion in university students. However, the vigor of university students increased significantly during forest contact in spring and early winter. In the study by Bielinis et al. [63], forest contact was clearly shown to increase the vigor of young adults more than the indoor environment did. However, in the present study we could not find a relationship between the color change and seasonal characteristics of the forest. Therefore, further studies are needed to more accurately gauge the effects of forest contact on confusion and vigor.

Biodiverse environments are associated with greater numbers of plant species and visual complexity [64]. Therefore, it was assumed that the lowest diversity would occur in spring, when the color profile is the most basic. Previous studies have shown that high levels of biodiversity contribute to positive psychological recovery [5,65,66,67]. However, in this study, psychological factors were all significantly positively affected across all K-POMS-B subscales in spring. In contrast to some of the previous studies outlining the merits of biodiversity, others have argued that moderate visual complexity is preferable to visual complexity for aesthetics and pleasure [68]. Therefore, future research should include forests with more visual differences and a longer monitoring period.

Contact with the forest significantly reduced fatigue in university students as compared to the indoor environment in each season recorded in this study. It is well documented that forest contact has a beneficial effect on people’s mood [69]; this study also showed that forest contact successfully reduced fatigue in university students regardless of the season. Reduction of fatigue is considered unrelated to the color changes of the forest, as fatigue was reduced only through actual contact with the forest.

Forest experience can provide health benefits through relaxation of the cardiovascular system [70,71], and may have the effect of lowering the activity of both the sympathetic and parasympathetic nervous systems after contact with the forest environment compared to the urban environment [38,69,72,73,74]. In this study, the high LF/HF ratio in spring indicates sympathetic nervous system activity, and the increase in the LF/HF ratio suggests that the forest experience, such as in spring, induces activation rather than relaxation of the autonomic nervous system. In addition, physiological characteristics were not observed in seasons other than spring. Short-term forest bathing does not affect changes in the autonomic nervous system [75], and changes in the autonomic nervous system are known to be affected by fatigue, physical constitution [76], gender [77], temperature [78], and recent experience [79]. Therefore, the results of this study are thought to be the result of various factors affecting the autonomic nervous system.

When the indoor and forest environments each season were compared through ANOVA analysis, fatigue was lower in the forest environment compared to the indoor environment, especially after participants experienced the forest environment in autumn, winter, and spring. Vigor was higher in the forest environment than in the indoor environment, especially after participants experienced the forest environment in spring.

Summarizing the above results, the environmental conditions of the spring forest alleviated the students’ tension, reduced fatigue, and increased their vigor. The characteristic of the spring forest environment investigated in this study was the simple distribution of green colors. Humans tend to feel comfortable in spaces with green plants [80,81]. Moreover, when the human emotional state is negative, the color green induces positive emotions and lowers negative emotions [82,83]. Viewing green plants in an indoor environment has been reported to be associated with greater stabilization of the autonomic nervous system and parasympathetic nerve activity as measured by HRV [84]. Therefore, it is thought that these green forest environmental conditions and temperature conditions, with an average of 17.9 °C, maximum of 23.9 °C, and minimum of 13.1 °C, in this study are more helpful in recovering university students’ mood states than the conditions of other survey days.

The effect of forest color and temperature on the moods and physiological states of university students was investigated in this study. Based on the study findings, it is believed that the forest color changes that occur each season had the effect of lowering the tension and anger in the study participants’ mood states. Human mood is understood to be affected by weather conditions, such as humidity, temperature, and hours of daylight [85], as well as by body temperature [86,87]. Therefore, future research should target these factors and include a longer period of monitoring and analysis to obtain clear results.

## 5. Conclusions

The spring forest color and temperature conditions are believed to have alleviated tension, reduced fatigue, and increased vigor in university students. Therefore, the results of this study can be used as a catalyst for planning urban forests for university campuses and their surrounding spaces. In addition, applying the methods used in this study to various age groups, classes, etc. will be helpful in creating urban forests that consider the characteristics of residents in each urban area.

## Figures and Tables

**Figure 1 ijerph-20-06338-f001:**
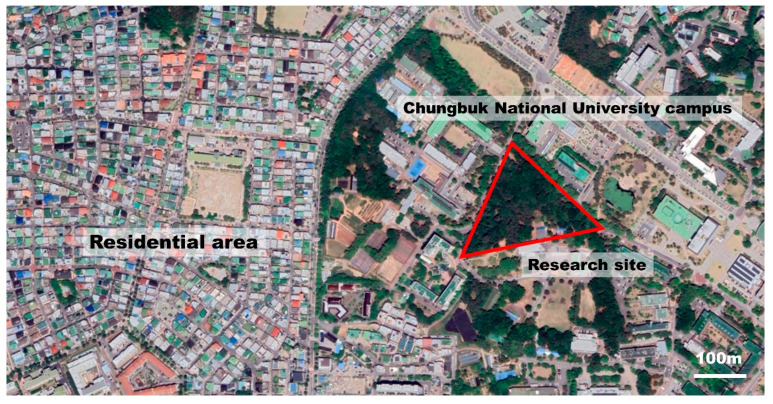
Location of the research site.

**Figure 2 ijerph-20-06338-f002:**
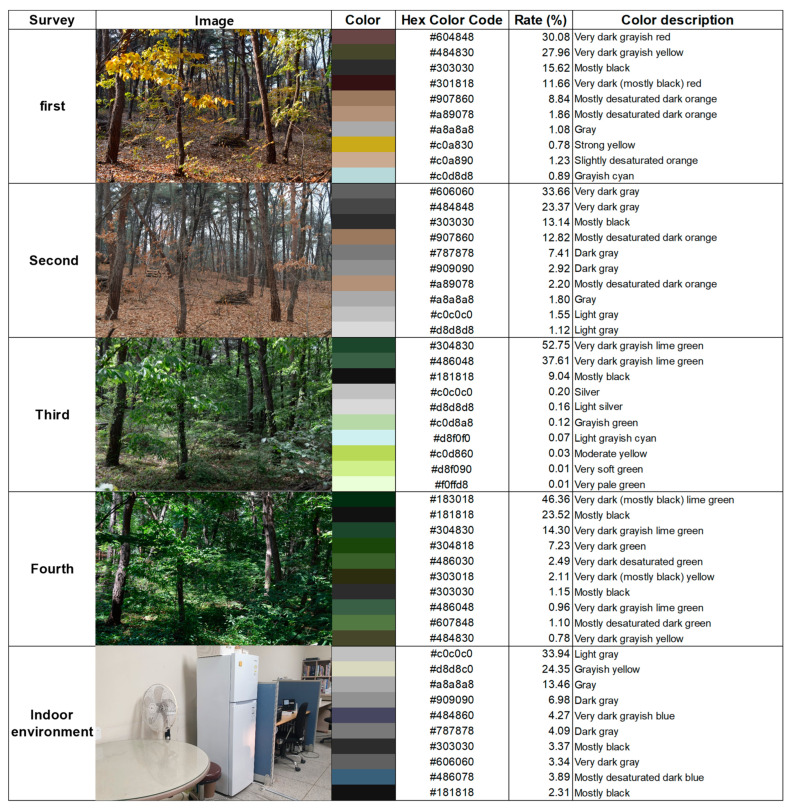
Distribution of color in the forest and indoor environments by survey day.

**Figure 3 ijerph-20-06338-f003:**
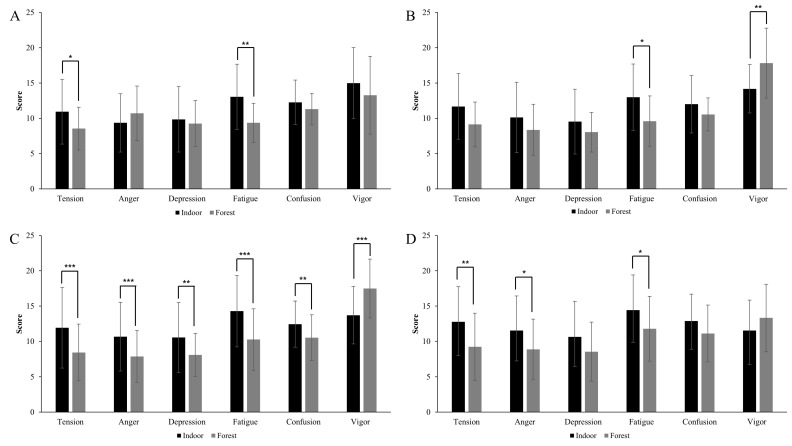
Difference in subscales of university students’ K-POMS-B for each survey day ((**A**): first survey day, (**B**): Second survey day, (**C**): third survey day, (**D**): fourth survey day, * *p* < 0.05, ** *p* < 0.01, *** *p* < 0.001).

**Table 1 ijerph-20-06338-t001:** Temperature for each survey day (°C).

	11 November 2021	12 December 2021	27 April 2022	15 June 2022
Average	10.8	6.1	17.9	19.6
Maximum	16.8	10.2	23.9	26.9
Minimum	4.9	2.7	13.1	14.8

**Table 2 ijerph-20-06338-t002:** Age and gender composition of study participants.

		First Survey Day	Second Survey Day	Third Survey Day	Fourth Survey Day
gender	male	8	7	12	13
	female	14	15	16	17
age	19	3	3	3	3
	20	7	6	8	8
	21	3	4	4	5
	22	3	3	6	6
	23	3	3	3	4
	24	2	2	3	3
	25	1	1	1	1
Total		22	22	28	30

**Table 3 ijerph-20-06338-t003:** Contents of the K-POMS-B.

	POMS-B Subscales					
	Tension	Anger	Depression	Fatigue	Confusion	Vigor
Item	Tense	angry	Sad	Bushed	Confused	Clear Headed
	Shaky	Grouchy	Unworthy	Fatigued	Muddled	Active
	On edge	Annoyed	Discouraged	Exhausted	Bewildered	Energetic
	Sympathetic	Resentful	Lonely	Sluggish	Efficient	Full of Pep
	Uneasy	peeved	Gloomy	Weary	Forgetful	Vigorous

**Table 4 ijerph-20-06338-t004:** ANOVA of the participation groups’ mood state for each survey day in indoor environment (Mean ± S.D.).

Variable	*n*	Tension	Anger	Depression	Fatigue	Confusion	Vigor
First survey	22	10.95 ± 4.59	9.36 ± 4.14	9.86 ± 4.64	13.05 ± 4.62	12.27 ± 3.15	15.00 ± 5.04
Second survey	22	11.68 ± 4.66	10.14 ± 4.97	9.55 ± 4.58	13.00 ± 4.70	12.00 ± 4.08	14.18 ± 3.43
Third survey	28	11.93 ± 5.68	10.68 ± 4.86	10.57 ± 4.93	14.29 ± 5.06	12.43 ± 3.27	13.71 ± 4.06
Fourth Survey	30	12.97 ± 5.08	11.53 ± 4.91	10.63 ± 5.05	14.43 ± 4.97	12.90 ± 3.81	11.53 ± 4.33
F		0.705	0.946	0.305	0.636	0.289	3.262 *
*p*		0.552	0.422	0.0822	0.593	0.833	0.025

* *p* < 0.05.

**Table 5 ijerph-20-06338-t005:** Differences in university students’ HRV subscales for each survey day.

Subscales	Survey Day	Treatment	*n*	M	S.D.	t	*p*
LF	First survey (11 November)	indoor	22	7.70	0.84	0.731	0.473
		forest		7.55	1.01		
	Second survey (12 December)	indoor	22	7.68	0.81	−0.218	0.829
		forest		7.70	0.70		
	Third survey (27 April)	indoor	28	7.68	0.81	−0.218	0.829
		forest		7.70	0.70		
	Fourth survey (15 June)	indoor	30	7.91	0.76	−0.226	0.823
		forest		7.94	0.75		
HF	First survey (11 November)	indoor	22	7.22	1.01	0.648	0.524
		forest		7.09	0.88		
	Second survey (12 December)	indoor	22	7.12	0.63	0.745	0.465
		forest		7.01	0.70		
	Third survey (27 April)	indoor	28	7.30	0.80	1.205	0.239
		forest		7.16	0.84		
	Fourth survey (15 June)	indoor	30	7.04	0.83	−1.292	0.207
		forest		7.19	0.67		
LF/HF	First survey (11 November)	indoor	22	1.09	0.15	0.535	0.598
		forest		1.07	0.10		
	Second survey (12 December)	indoor	22	1.07	0.11	−1.402	0.176
		forest		1.11	0.10		
	Third survey (27 April)	indoor	28	1.09	0.11	−2.073 *	0.048
		forest		1.13	0.10		
	Fourth survey (15 June)	indoor	30	1.13	0.02	1.884	0.070
		forest		1.09	0.01		

* *p* < 0.05.

**Table 6 ijerph-20-06338-t006:** Differences between mood state sub-factors in indoor and seasonal forest environments.

Subscales	Environment Conditions and Survey Days	n	Mean	S.D.	F	*p*	Post-Verification (Bonferroni)
Tension	Indoor ^a^ (a)	102	11.91	5.01	6.043 ***	0.000	b, d, e < a
	First ^b^ (b)	22	8.55	3.04			
	Second ^c^ (c)	22	9.14	3.18			
	Third ^d^ (d)	28	8.46	4.00			
	Fourth ^e^ (e)	30	9.23	4.75			
Anger	Indoor (a)	102	10.53	4.75	3.236 *	0.013	
	First (b)	22	10.73	3.87			
	Second (c)	22	8.36	3.63			
	Third (d)	28	7.89	3.65			
	Fourth (e)	30	8.87	4.29			
Depression	Indoor (a)	102	10.22	4.78	2.531 *	0.042	
	First (b)	22	9.27	3.25			
	Second (c)	22	8.05	2.80			
	Third (d)	28	8.11	3.03			
	Fourth (e)	30	8.57	4.17			
Fatigue	Indoor (a)	102	13.78	4.84	8.733 ***	0.000	b, c, d < a
	First (b)	22	9.36	2.77			
	Second (c)	22	9.59	3.59			
	Third (d)	28	10.29	4.34			
	Fourth (e)	30	11.80	4.57			
Confusion	Indoor (a)	102	12.44	3.56	3.004 *	0.020	
	First (b)	22	11.32	2.23			
	Second (c)	22	10.55	2.34			
	Third (d)	28	10.54	3.23			
	Fourth (e)	30	11.13	4.02			
Vigor	Indoor (a)	102	13.45	4.39	7.982 ***	0.000	c, d > a
	First (b)	22	13.27	5.52			
	Second (c)	22	17.82	4.95			
	Third (d)	28	17.50	4.18			
	Fourth (e)	30	13.33	4.78			

* *p* < 0.05, *** *p*< 0.001. ^a^: indoor environment, ^b^: forest environment at first survey day, ^c^: forest environment at second survey day, ^d^: forest environment at third survey day, ^e^: forest environment at fourth survey day.

**Table 7 ijerph-20-06338-t007:** Differences between HRV sub-factors in indoor and seasonal forest environments.

Subscales	Survey Day	n	Mean	S.D.	F	*p*
LF	Indoor ^a^	102	7.81	0.77	1.197	0.313
	First ^b^	22	7.55	1.01		
	Second ^c^	22	7.70	0.70		
	Third ^d^	28	7.98	0.85		
	Fourth ^e^	30	7.94	0.75		
HF	Indoor	102	7.17	0.82	0.234	0.919
	First	22	7.09	0.88		
	Second	22	7.01	0.70		
	Third	28	7.16	0.84		
	Fourth	30	7.19	0.67		
LF/HF	Indoor	102	1.10	0.12	1.070	0.373
	First	22	1.07	0.10		
	Second	22	1.11	0.10		
	Third	28	1.13	0.10		
	Fourth	30	1.09	0.07		

^a^: indoor environment, ^b^: forest environment at first survey day, ^c^: forest environment at second survey day, ^d^: forest environment at third survey day, ^e^: forest environment at fourth survey day.

## Data Availability

All data are available within the paper.

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
