# Peer review of "Seasonal Forest Changes of Color and Temperature: Effects on the Mood and Physiological State of University Students"

_ijerph, 2023, doi:10.3390/ijerph20146338_

Round 1
Reviewer 1 Report
In this study, the authors attempted to analyze the impact of color variations over time in forest environments on the emotional state of college students. The paper extensively described the increasing emotional disorders among contemporary college students, and provided the research background on the relaxation and stress reduction effects of natural landscapes, explaining the significance of studying forest colors and their importance for the emotional state of college students. The investigation was conducted in four phases, describing the data collection process and quantifying the specific seasonal changes in the forest. After capturing photographs of the forest, the color visual information of the forest environment for each survey day was analyzed through color extraction, and participants' emotions were measured using the K-POMS-B scale. The results of this study can provide valuable data that contribute to considering the role of color in urban forest planning for universities and other public spaces. However, there are some problems with the current manuscript, and the specific comments are shown below.
1. This present study employed questionnaire measurements, but this self-report assessment may not fully explain the health benefits of the forest environment, as it is susceptible to participants' subjective bias. The authors should consider combining physiological measurements, such as heart rate, blood pressure, heart rate variability, skin conductance level, or electroencephalogram.
2. There is no explanation for the experimental design method used in the study. It was a within-subjects design, where each participant took part in four surveys, or a between-subjects design, where different participants were used for each survey? This involved the selection of statistical method. If it was a within-subjects design, the repeated measures ANOVA should be chosen.
3. The Materials and Methods section lacks basic background information on the age, height, weight, caffeine intake, and medical history (especially mental illness) of the college students. It is recommended to include a description of the demographic information of the participating college students to provide a better understanding of the study population.
4. It is suggested to analyze and discuss whether there are gender differences in forest color perception by including a discussion or analysis of potential differences between male and female college students in each experimental round. This would provide a more comprehensive discussion of the experimental results.
5. The Materials and Methods section should include information about the time range of the experiments conducted each day. Specifically, it should specify the start and end times of the daily experiments and whether the experiment duration was relatively fixed in the morning or afternoon, taking into account variations in daylight, such as avoiding the noon sun. This information is important for understanding the temporal aspects of the study design and potential influences of diurnal variations.
6. In the description of the experimental environment, specifically the "Temperature" section, the author did not provide basic information about the indoor temperature. While the temperature changes in the forest during different seasons were described, the data analysis included a paired t-test comparing indoor and forest conditions. It is recommended to provide additional information about the indoor temperature, as this would enhance the coherence of the context. Including basic information about the indoor temperature will help readers understand the temperature conditions experienced by the participants and strengthen the connection between the experimental setup and the data analysis.
7. Except for temperature, environmental factors such as humidity and wind speed can also affect emotional states. This study only counted the temperature on each survey day and lacked other environmental data. It is recommended to supplement.
8. In the line 115 "After adapting to the laboratory environment," it is not clear how the participants' adaptation to the laboratory environment was determined. It would be beneficial for the authors to provide more information on the criteria used to assess participants' adaptation. Additionally, it would be helpful to clarify whether the participants spent the same duration of time (e.g., 10-20 minutes) in both the indoor and forest environments to ensure a consistent comparison between the two conditions. Providing such details will enhance the clarity and replicability of the study methodology.
9. The present study only analyzed the differences between forest environment and indoor environment during each survey day, lacking direct comparison of the impact of forest environment of different seasons on emotional state.
10. In the line 308-309, “Furthermore, it was confirmed that the forest environment conditions in the spring season had a positive association with all of the subscales of POMS-B for the study participants.” In the experimental results, there was no analysis of the correlation between forest environment conditions and POMS-B. The reviewer understands that "environment conditions" should include factors such as temperature, humidity, air composition, and air ions. If the author is referring to the relationship between color changes and emotional states, it is advised to consider modifying the wording accordingly.
11. Since the author is discussing the relationship between color changes and emotional changes, it is recommended to analyze the correlation between color proportions and emotions. If there is a significant correlation between environments with a high proportion of green color and positive emotions, it would provide more comprehensive results when discussing the impact of color changes on emotions.
12. Table 4 spans multiple pages and appears to be quite lengthy. It is recommended to consider alternative ways of presenting the results, such as using figures to illustrate the differences. If the author believes that the data in the table is crucial, it could be included as supplementary material to ensure its availability for interested readers.
13. Please mention the limitations of this study and the direction of future research in the last paragraph of the discussion.
The English language of this manuscript needs further improvement.
Author Response
Dr.
Editor-in-Chief
International Journal of Environmental Research and Public Health
Dear Sir,
Thank you for considering our submitted manuscript ID ijerph-2458177.
Indeed we are grateful for your invaluable comments on our manuscript entitled “Seasonal forest color changes: Effects on the mood state of university students". As per the reviewer’s suggestion, I have revised the manuscript thoroughly. Herewith I have given our response to reviewer comments and revised the manuscript by using red color text. I have assured you that the manuscript is now devoid of any mistake. With this I hope the paper is now changed as per your direction and I hope to get a positive answer from you soon
Thank you Sir.
Sincerely,
Hwayong Lee
Department of forest science, Chungbuk National University
Chungdae-ro 1, Seowon-Gu, Cheongju, Chungbuk 28644, Korea
+82-43-261-2537
+82-43-272-5921
leehy@chungbuk.ac.kr
Reviewer 1
In this study, the authors attempted to analyze the impact of color variations over time in forest environments on the emotional state of college students. The paper extensively described the increasing emotional disorders among contemporary college students, and provided the research background on the relaxation and stress reduction effects of natural landscapes, explaining the significance of studying forest colors and their importance for the emotional state of college students. The investigation was conducted in four phases, describing the data collection process and quantifying the specific seasonal changes in the forest. After capturing photographs of the forest, the color visual information of the forest environment for each survey day was analyzed through color extraction, and participants' emotions were measured using the K-POMS-B scale. The results of this study can provide valuable data that contribute to considering the role of color in urban forest planning for universities and other public spaces. However, there are some problems with the current manuscript, and the specific comments are shown below.
Responses the Reviewer comment: Thank you for your comments. We accepted your suggestion and revised the manuscript.
- This present study employed questionnaire measurements, but this self-report assessment may not fully explain the health benefits of the forest environment, as it is susceptible to participants' subjective bias. The authors should consider combining physiological measurements, such as heart rate, blood pressure, heart rate variability, skin conductance level, or electroencephalogram.
Responses the Reviewer comment: We accepted your suggestion and added physiological data related to HRV that were not written in the original manuscript.
- There is no explanation for the experimental design method used in the study. It was a within-subjects design, where each participant took part in four surveys, or a between-subjects design, where different participants were used for each survey? This involved the selection of statistical method. If it was a within-subjects design, the repeated measures ANOVA should be chosen.
Responses the Reviewer comment: In this study, a group with the same characteristics was recruited for each survey day, and the study was conducted, but no consecutive experiments were conducted with the same person. This information was briefly added to materials and methods in manuscript.
- The Materials and Methods section lacks basic background information on the age, height, weight, caffeine intake, and medical history (especially mental illness) of the college students. It is recommended to include a description of the demographic information of the participating college students to provide a better understanding of the study population.
Responses the Reviewer comment: At the time of the study, the participants had no major physical or mental problems. This content was briefly written, and we did not write much information about students, but we did research about age, so we added it.
- It is suggested to analyze and discuss whether there are gender differences in forest color perception by including a discussion or analysis of potential differences between male and female college students in each experimental round. This would provide a more comprehensive discussion of the experimental results.
Responses the Reviewer comment: In this study, we tried to analyze university students in a large category without distinguishing between men and women. In addition, the number of sample groups was small to distinguish between men and women, so it was difficult to infer population information, so it was not included in the manuscript. However, considering the reviewer's suggestion, we would like to attach gender-separated statistical data as supplementary data.
- The Materials and Methods section should include information about the time range of the experiments conducted each day. Specifically, it should specify the start and end times of the daily experiments and whether the experiment duration was relatively fixed in the morning or afternoon, taking into account variations in daylight, such as avoiding the noon sun. This information is important for understanding the temporal aspects of the study design and potential influences of diurnal variations.
Responses the Reviewer comment: The experiment was performed from 11:00 to 15:00 on a sunny day. This content has been added to the manuscript.
- In the description of the experimental environment, specifically the "Temperature" section, the author did not provide basic information about the indoor temperature. While the temperature changes in the forest during different seasons were described, the data analysis included a paired t-test comparing indoor and forest conditions. It is recommended to provide additional information about the indoor temperature, as this would enhance the coherence of the context. Including basic information about the indoor temperature will help readers understand the temperature conditions experienced by the participants and strengthen the connection between the experimental setup and the data analysis.
Responses the Reviewer comment: Thank you for this valuable comment. In this study, the indoor temperature is controlled around 25 °C throughout the year. This content has been added to the manuscript.
- Except for temperature, environmental factors such as humidity and wind speed can also affect emotional states. This study only counted the temperature on each survey day and lacked other environmental data. It is recommended to supplement.
Responses the Reviewer comment: The data of the Korea Meteorological Administration we used did not include information such as humidity and etc., so we could not write it in the manuscript. Accordingly, the title and purpose of the study were revised by limiting it to the color and temperature of the forest, and the need for further study considering various factors was written in the discussion and conclusion sections.
- In the line 115 "After adapting to the laboratory environment," it is not clear how the participants' adaptation to the laboratory environment was determined. It would be beneficial for the authors to provide more information on the criteria used to assess participants' adaptation. Additionally, it would be helpful to clarify whether the participants spent the same duration of time (e.g., 10-20 minutes) in both the indoor and forest environments to ensure a consistent comparison between the two conditions. Providing such details will enhance the clarity and replicability of the study methodology.
Responses the Reviewer comment: In this study, participants were tested in indoor and forest environments for the same amount of time (10-20 minute). This content has been revised in the method of manuscript.
- The present study only analyzed the differences between forest environment and indoor environment during each survey day, lacking direct comparison of the impact of forest environment of different seasons on emotional state.
Responses the Reviewer comment: Thank you for your comment. We accepted your suggestion and analyzed the difference between the indoor environment and each season using ANOVA and Bonferroni correction and added it to the manuscript.
- In the line 308-309, “Furthermore, it was confirmed that the forest environment conditions in the spring season had a positive association with all of the subscales of POMS-B for the study participants.” In the experimental results, there was no analysis of the correlation between forest environment conditions and POMS-B. The reviewer understands that "environment conditions" should include factors such as temperature, humidity, air composition, and air ions. If the author is referring to the relationship between color changes and emotional states, it is advised to consider modifying the wording accordingly.
Responses the Reviewer comment: This part is our mistake. We have revised the conclusion part of your comment.
- Since the author is discussing the relationship between color changes and emotional changes, it is recommended to analyze the correlation between color proportions and emotions. If there is a significant correlation between environments with a high proportion of green color and positive emotions, it would provide more comprehensive results when discussing the impact of color changes on emotions.
Responses the Reviewer comment: Thank you for your comments. We analyzed the correlation, but found a very low correlation. Correspondingly, it is believed that correlation analysis requires a longer period of time to conduct this study under different environmental conditions. Additionally, other factors in the forest are believed to influence emotions. Instead of correlation analysis, we additionally wrote about the effect of green on human mood and physiological state in the discussion section.
- Table 4 spans multiple pages and appears to be quite lengthy. It is recommended to consider alternative ways of presenting the results, such as using figures to illustrate the differences. If the author believes that the data in the table is crucial, it could be included as supplementary material to ensure its availability for interested readers.
Responses the Reviewer comment: The table has been converted into a picture. And the table was created as a supplementary table.
- Please mention the limitations of this study and the direction of future research in the last paragraph of the discussion.
Responses the Reviewer comment: At the end of the discussion, using cited literature, various factors affecting human mood were written, and future research was briefly written.

Reviewer 2 Report
The study doesn’t really test colour. It tests forest cf indoor at 4 seasons. Table 4 includes 24 tests, so with a bonferroni correction, p<0.002 for significance (0.05/24). So, all subscales show forest different from indoor ONLY for spring. So the title and article should be rewritten to say, mood effects of forest are significant only in spring. That’s very publishable, but it needs a rewrite. Can mention colour, but there’s no evidence that colour drives the effect. Could be many different factors.
Author Response
Dr.
Editor-in-Chief
International Journal of Environmental Research and Public Health
Dear Sir,
Thank you for considering our submitted manuscript ID ijerph-2458177.
Indeed we are grateful for your invaluable comments on our manuscript entitled “Seasonal forest color changes: Effects on the mood state of university students". As per the reviewer’s suggestion, I have revised the manuscript thoroughly. Herewith I have given our response to reviewer comments and revised the manuscript by using red color text. I have assured you that the manuscript is now devoid of any mistake. With this I hope the paper is now changed as per your direction and I hope to get a positive answer from you soon
Thank you Sir.
Sincerely,
Hwayong Lee
Department of forest science, Chungbuk National University
Chungdae-ro 1, Seowon-Gu, Cheongju, Chungbuk 28644, Korea
+82-43-261-2537
+82-43-272-5921
leehy@chungbuk.ac.kr
Reviewer 2
The study doesn’t really test colour. It tests forest cf indoor at 4 seasons. Table 4 includes 24 tests, so with a bonferroni correction, p<0.002 for significance (0.05/24). So, all subscales show forest different from indoor ONLY for spring. So the title and article should be rewritten to say, mood effects of forest are significant only in spring. That’s very publishable, but it needs a rewrite. Can mention colour, but there’s no evidence that colour drives the effect. Could be many different factors.
Responses the Reviewer comment: Thank you for your comments. We accepted your suggestion and revised the manuscript. We attempted a reanalysis using Bonferroni correction, and wrote a revised manuscript in consideration of comments from other reviewers. Since the color of the forest is one of many factors that can affect the mood state, it was confirmed that further research is needed. Accordingly, the title and purpose of the study were limited, and the final part of the overall review was rewritten in the manuscript that further research is needed because the color is one of many factors.
Round 2
Reviewer 1 Report
The authors made meticulous revisions to the paper, adding essential experimental information to the manuscript's Materials and Methods section, such as the duration of participants' exposure to indoor and forest environments. They also included previously unmentioned physiological data related to HRV (heart rate variability) in the revised version. The title and objectives of the study were modified to specifically focus on forest color and temperature. Furthermore, in the Discussion and Conclusion sections, the authors highlighted the necessity of further research considering various factors. However, there are some issues with the current manuscript. Please provide specific comments for further evaluation.
1. In the section of Materials and Methods, there is a lack of information on the HRV indicators used in the study, such as the full names of LF, HF, and LF/HF, and the meanings when they are raised or lowered.
2. Line 255-256: in the post-test, the forest environment experience was found in the indoor environment showed higher ‘vigor’. However, the change in vigor scores were not consistent across the four forest environments. At the first and forth survey day, the vigor scores in the forest environment were lower than that in the indoor environment. While at the second and third survey day, the vigor scores in the forest environment were higher than that in the indoor environment (Table 6).
3. In the section of Discussion, there is little content related to physiological indicators, i.e. HRV, and it is recommended to increase the relevant discussion.
Minor editing of English language required
Author Response
Dr.
Editor-in-Chief
International Journal of Environmental Research and Public Health
Dear Sir,
Thank you for considering our submitted manuscript ID ijerph-2458177.
Indeed we are grateful for your invaluable comments on our manuscript entitled “Seasonal forest changes of color and temperature: Effects on the mood and physiological state of university students". As per the reviewer’s suggestion, I have revised the manuscript thoroughly. Herewith I have given our response to reviewer comments and revised the manuscript by using red color text. We have assured you that the manuscript is now devoid of any mistake. With this I hope the paper is now changed as per your direction and I hope to get a positive answer from you soon
Thank you Sir.
Sincerely,
Hwayong Lee
Department of forest science, Chungbuk National University
Chungdae-ro 1, Seowon-Gu, Cheongju, Chungbuk 28644, Korea
+82-43-261-2537
+82-43-272-5921
leehy@chungbuk.ac.kr
Reviewer 1
The authors made meticulous revisions to the paper, adding essential experimental information to the manuscript's Materials and Methods section, such as the duration of participants' exposure to indoor and forest environments. They also included previously unmentioned physiological data related to HRV (heart rate variability) in the revised version. The title and objectives of the study were modified to specifically focus on forest color and temperature. Furthermore, in the Discussion and Conclusion sections, the authors highlighted the necessity of further research considering various factors. However, there are some issues with the current manuscript. Please provide specific comments for further evaluation.
Responses the Reviewer comment: Thank you for your comments. We accepted your suggestion and revised the manuscript.
- In the section of Materials and Methods, there is a lack of information on the HRV indicators used in the study, such as the full names of LF, HF, and LF/HF, and the meanings when they are raised or lowered.
Responses the Reviewer comment: We have added the contents of your comments to the materials and methods part in the manuscript.
- Line 255-256: in the post-test, the forest environment experience was found in the indoor environment showed higher ‘vigor’. However, the change in vigor scores were not consistent across the four forest environments. At the first and forth survey day, the vigor scores in the forest environment were lower than that in the indoor environment. While at the second and third survey day, the vigor scores in the forest environment were higher than that in the indoor environment (Table 6).
Responses the Reviewer comment: Thank you for your valuable comments. The part you commented on was written incorrectly. This part has been completely rewritten.
- In the section of Discussion, there is little content related to physiological indicators, i.e. HRV, and it is recommended to increase the relevant discussion.
Responses the Reviewer comment: We accepted the reviewer's suggestion and added content related to HRV in the discussion part.
Comments on the Quality of English Language
Minor editing of English language required
Responses the Reviewer comment: The manuscript was revised in response to the reviewer's comments. And the revised manuscript was corrected using a professional English proofreading service. The editing certificate of this manuscript is attached.
